# OPtimising Treatment for MIld Systolic hypertension in the Elderly (OPTiMISE): protocol for a randomised controlled non-inferiority trial

James P Sheppard,[1] Jenni Burt,[2] Mark Lown,[3] Eleanor Temple,[1] John Benson,[4] Gary A Ford,[1] Carl Heneghan,[1] F D Richard Hobbs,[1] Sue Jowett,[5] Paul Little,[3] Jonathan Mant,[4] Jill Mollison,[1] Alecia Nickless,[1] Emma Ogburn,[1] Rupert Payne,[6] Marney Williams,[7] Ly-Mee Yu,[1] Richard J McManus[1]

For numbered affiliations see end of article.

**Correspondence to**
Dr James P Sheppard;
james.sheppard@phc.ox.ac.uk

## ABSTRACT

**Introduction** Recent evidence suggests that larger blood pressure reductions and multiple antihypertensive drugs may be harmful in older people, particularly frail individuals with polypharmacy and multimorbidity. However, there is a lack of evidence to support deprescribing of antihypertensives, which limits the practice of medication reduction in routine clinical care. The aim of this trial is to examine whether antihypertensive medication reduction is possible in older patients without significant changes in blood pressure control at follow-up.

**Methods and analysis** This trial will use a primary care-based, open-label, randomised controlled trial design. A total of 540 participants will be recruited, aged ≥80 years, with systolic blood pressure <150 mm Hg and receiving ≥2 antihypertensive medications. Participants will have no compelling indication for medication continuation and will be considered to potentially benefit from medication reduction due to existing polypharmacy, comorbidity and frailty. Following a baseline appointment, individuals will be randomised to a strategy of medication reduction (intervention) with optional self-monitoring or usual care (control). Those in the intervention group will have one antihypertensive medication stopped. The primary outcome will be to determine if a reduction in medication can achieve a proportion of participants with clinically safe blood pressure levels at 12-week follow-up (defined as a systolic blood pressure <150 mm Hg), which is non-inferior (within 10%) to that achieved by the usual care group. Qualitative interviews will be used to understand the barriers and facilitators to medication reduction. The study will use economic modelling to predict the long-term effects of any observed changes in blood pressure and quality of life.

**Ethics and dissemination** The protocol, informed consent form, participant information sheet and all other participant facing material have been approved by the Research Ethics Committee (South Central—Oxford A; ref 16/SC/0628), Medicines and Healthcare products Regulatory Agency (ref 21584/0371/001–0001), host institution(s) and Health Research Authority. All research outputs will be published in peer-reviewed journals and presented at national and international conferences.

### Strengths and limitations of this study

► This will be the first UK randomised controlled trial to compare a strategy of antihypertensive medication reduction with usual care in primary care.
► The pragmatic trial design, with broad inclusion criteria, will make findings of the study externally valid in routine clinical practice.
► Allowing the attending general practitioner to choose the medication to be reduced will maximise external validity of the trial results, but precludes the possibility of blinding the participants and investigators.
► The trial will be powered to detect a non-inferior difference in blood pressure control at follow-up, but not necessarily secondary outcomes such as differences in rates of cardiovascular disease, adverse events and quality of life.

**Trial registration number** EudraCT 2016-004236-38; ISRCTN97503221; Pre-results.

## INTRODUCTION

The general population is ageing[1] and, consequently, the number of people living with age-related chronic conditions is increasing.[2] Hypertension is the number one comorbid condition in older people with multiple chronic conditions[3] and 52% of those aged ≥80 years are prescribed two or more antihypertensive drugs (equivalent to approximately 1.25 million people in the UK).[4] Blood pressure lowering has been shown to be effective at preventing stroke and cardiovascular disease in healthy individuals aged ≥80 years with stage 2 hypertension (systolic blood pressure of >160 mm Hg) and high-risk stage 1 hypertension.[5][6] However, as with many trials,[7][8] these studies included healthier populations with lower polypharmacy and multimorbidity than might be expected in the general elderly population.

In addition, there is evidence to suggest that larger blood pressure reductions and multiple antihypertensive prescriptions may be harmful in older people.[9][10] Evidence from observational studies also suggests that higher intensity blood pressure treatment is associated with increased risk of falls in older people,[11] although this is also disputed.[5]

Some patients consider the increased risk of falls and other adverse events (AEs) to be as important as the risk of myocardial infarction or stroke, particularly those taking medications for primary prevention of cardiovascular disease.[12] Thus, decisions over blood pressure lowering in the elderly, particularly the frail elderly, require the weighing of harms and quality of life. However, clinicians can often struggle to stop prescribing medication due to a perceived lack of evidence, fear of the reaction of other prescribers, fear of precipitating events such as stroke or angina and concern that patients will feel their care is being cut.[13][14]

There is limited evidence from randomised trials examining the safety of antihypertensive medication reduction or withdrawal.[15–19] The Hypertension in the Very Elderly Trial (HYVET)[5] enrolled some patients on antihypertensive treatment who were then randomised to placebo (effectively complete medication withdrawal) and the second Australian National Blood Pressure trial investigators followed up participants who withdrew medication during the trial run-in period but who were not randomised into the trial. They found younger patients with lower baseline blood pressure and fewer drug prescriptions were more likely to sustain medication withdrawal at 12-month follow-up.[20][21] However, there are few trials comparing a specified strategy of antihypertensive medication reduction with usual care in terms of effects on blood pressure control and quality of life.[17] In addition, there are no previous economic modelling studies of a strategy of medication reduction in the elderly.

The aim of this work will be to examine whether antihypertensive medication reduction in patients with controlled systolic hypertension (≤150 mm Hg) who are being prescribed two or more antihypertensives is possible without significant changes in blood pressure control at follow-up.

## METHODS
### Aims and outcomes
The aim of this study is to determine whether antihypertensive medications can be safely reduced without systolic blood pressure increasing beyond what is clinical acceptable at follow-up.

The primary outcome is the proportion of participants with clinically acceptable blood pressure levels at 12-week follow-up (defined as a systolic blood pressure <150 mm Hg). Secondary outcomes will examine:

▶ The proportion of participants in the intervention arm who maintain medication reduction through to follow-up (ie, are *not* restarted on therapy).

▶ The difference in quality of life (according to EQ-5D-5L) between groups at 12-week follow-up.
▶ The difference in frailty (according to the frailty index)[22] between the two groups at 12-week follow-up.
▶ The difference in the change in mean clinic systolic blood pressure (from baseline) between the two groups at 12-week follow-up.
▶ The difference in reported side effects to medication between the two groups at 12-week follow-up (including coughs, dizziness, syncope, and ankle swelling).
▶ The difference in routinely reported serious adverse events (SAEs) between the two groups at 12-week follow-up (hospitalisation due to falls, myocardial infarction, stroke or all-cause mortality).

### Design
This trial will use a primary care-based, open-label, randomised controlled, two-parallel groups, non-inferiority trial design, recruiting 540 participants with controlled blood pressure (systolic <150 mm Hg) on two or more antihypertensive treatments. Participants will be randomised to a strategy of medication reduction (intervention) or usual care (control) and followed-up for 12 weeks (figure 1). Embedded qualitative and economic analyses will examine barriers and facilitators to medication reduction and the cost-effectiveness of the approach.

### Trial participants
Patients eligible for the trial will be aged ≥80 years, with systolic blood pressure <150 mm Hg (current UK guideline recommendation)[23] receiving ≥2 antihypertensive medications. They will have no compelling indication for medication continuation and in the opinion of the attending GP, may potentially benefit from medication reduction due to existing polypharmacy, comorbidity and/or frailty (box 1).

Participants will be identified and recruited from general practices via the UK Clinical Research Network. Potentially eligible patients will be identified by trained practice staff searching practice-based electronic disease registers using a standardised strategy. GPs will be asked to check the search results and remove people whom they believe to be unsuitable to participate in the study. Remaining potentially eligible patients will be sent letters of invitation from their GP and those expressing an interest in the trial will be asked to attend a screening and baseline appointment. Patients not responding to the first invitation will receive one reminder letter (up to 4 weeks after the first letter). Other potentially eligible patients may also be approached opportunistically by a member of the clinical care team. Those who do not wish to take part will be asked to fill in a short questionnaire detailing their reasons.

### Baseline visit
Eligible patients will have informed consent taken by the GP. During the consent appointment, the GP will show a 2 min study video infographic (see online supplementary

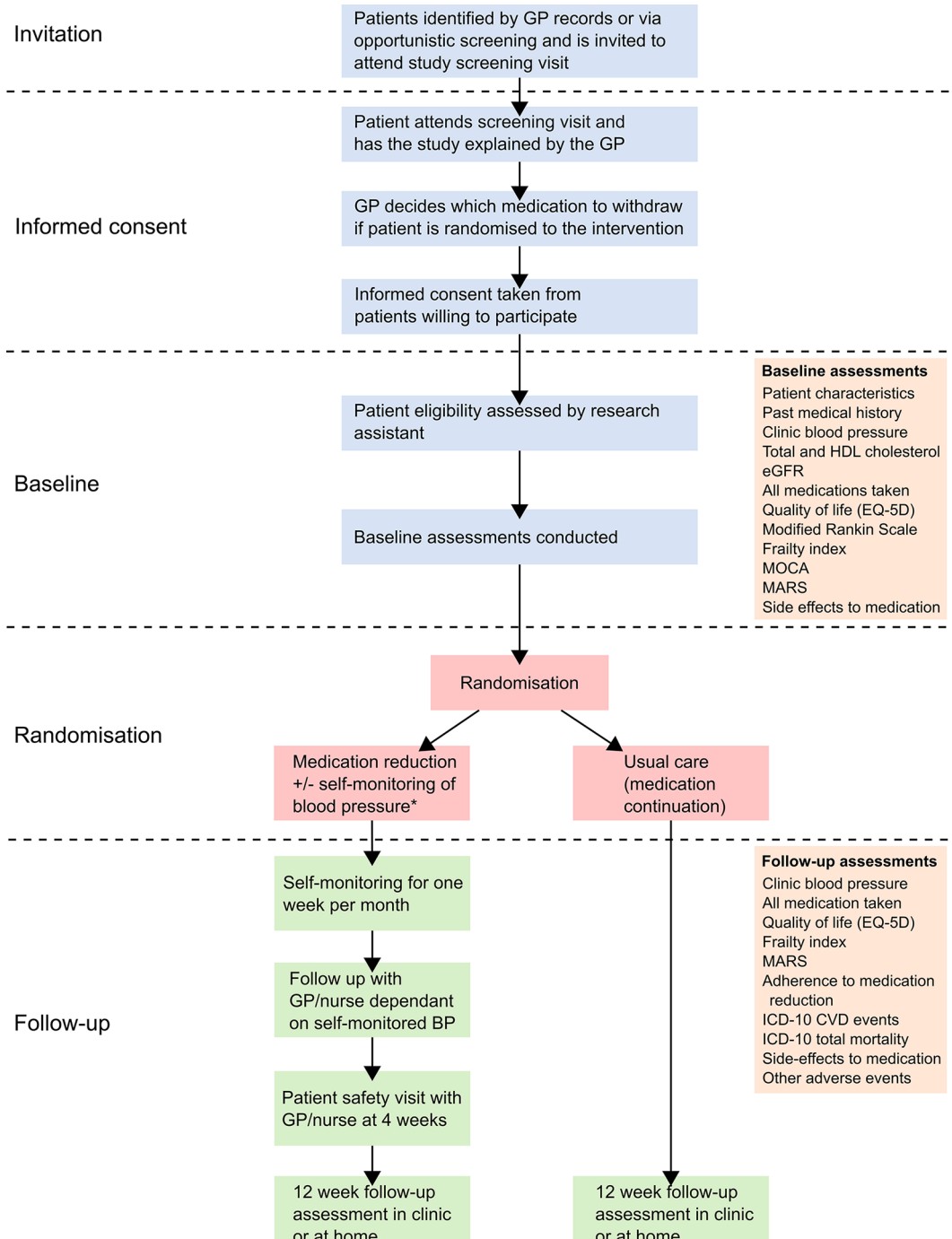

**Figure 1** Trial flow diagram. *Monitoring of blood pressure at home will be encouraged but those not willing or able will still be included in the trial. All participants will be asked to attend a safety monitoring visit with their GP/nurse 4 weeks after baseline. GP, general practitioner; BP, blood pressure; HDL, high-density lipoprotein; ICD, International Statistical Classification of Diseases and Related Health Problems; CVD, cardiovascular disease; eGFR, estimated glomerular filtration rate; MARS, Medication Adherence Rating Scale; MOCA, Montreal Cognitive Assessment.

material) and go through the participant information sheet explaining the exact nature of the trial. Having had a chance to ask questions, those individuals willing to participate will give written informed consent by means of a participant dated signature and dated signature of the GP who presented and obtained the informed consent.

Some participants will be invited to have their interview audio-recorded for qualitative analysis during their study visits. Those who are interested will be asked to sign a response slip prior to meeting the GP. Consent to audio-recordings will not have a bearing on an individual's care or eligibility for the main trial.

Those giving informed consent will be screened using the criteria in box 1 and undergo baseline measurements and randomisation by a member of the research team via participant questionnaires and a detailed notes review (table 1).

**Inclusion criteria**
- ► Participant is willing and able to give informed consent for participation in the trial.
- ► Male or female, aged 80 years or above.
- ► Clinic systolic blood pressure <150 mm Hg (according to screening measurement at baseline—clinic blood pressure defined as the mean of the second and third readings taken at 1 min intervals).
- ► Prescribed two or more antihypertensive medications to lower blood pressure for at least 12 months prior to trial entry. Antihypertensive medications defined as any ACE inhibitor, angiotensin II receptor blocker, calcium channel blocker, thiazide and thiazide-like diuretic, potassium-sparing diuretic, alpha-blocker, beta-blocker, vasodilator antihypertensives, centrally acting antihypertensives, direct renin inhibitors, adrenergic neuron blocking drugs or loop diuretics.
- ► Stable dose of antihypertensive medications for at least 4 weeks prior to trial entry.
- ► In the investigator's opinion, could potentially benefit from medication reduction due to existing polypharmacy, comorbidity, non-adherence or dislike of medicines and/or frailty.
- ► In the investigator's opinion, is able and willing to comply with all trial requirements.

**Exclusion criteria**
- ► A participant has heart failure due to LVSD and is on only ACE inhibitors/ARBs and/or beta-blockers and/or spironolactone (removing any of which would be contraindicated).
- ► A participant has heart failure but has not had an echocardiogram since its onset (might have undiagnosed LVSD and a compelling need for ACE inhibitors/ARB and beta-blockers).
- ► Investigator deems that there is a compelling indication for antihypertensive medication continuation.
- ► Any other significant disease or disorder which, in the opinion of the investigator, may either put the participants at risk because of participation in the trial, or may influence the result of the trial, or the participant's ability to participate in the trial (eg, terminal illness, house bound and unable to attend baseline and follow-up clinics).
- ► Suffered a myocardial infarction or stroke within the past 12 months.
- ► Blood pressure being managed outside of primary care.
- ► Unable to provide consent due to incapacity.
- ► A participant with secondary hypertension or previous accelerated or malignant hypertension.
- ► Participants who have participated in another research trial involving antihypertensive medication in the past 4 weeks.

ARB, angiotensin II receptor blocker; LVSD, left ventricular systolic dysfunction.

Blood pressure will be measured in a standardised fashion using the clinically validated[24] BpTRU blood pressure monitor, which automatically records six blood pressure measurements at 1 min intervals. Blood pressure readings will be taken in the left arm (where appropriate) after participants have been seated for at least 5 min of rest, using an appropriate sized cuff. The mean of the second and third readings will be used to define the primary outcome. To test for orthostatic hypotension, two further readings will be taken in the standing position after 1 and 3 min.[25] Only the research facilitator/nurse will be present during the blood pressure measurements. Orthostatic hypotension will be defined as a >20 mm Hg drop in systolic blood pressure within 3 min of standing.

All data will be collected via an electronic case report form (eCRF) linked to the study database. Participants will be given the option to enter responses to questionnaires themselves or with assistance from the research team. Where questionnaires are not validated for use on a tablet computer,[26] or where individuals are not comfortable using one, paper copies will be made available for completion.

## Randomisation

Consenting participants will be individually randomised (1:1 allocation ratio) to one of two study arms using a fully validated web-based system (Sortition) with manual telephone back up. Participants will not be randomised until after consent has been taken and baseline assessments have been completed. A computer-generated non-deterministic algorithm, minimising on practice and baseline systolic blood pressure will be used to ensure these covariates are balanced between the two intervention arms.

The study will use an open-label design, so patients and practitioners will not be blinded to the intervention or study end points. Therefore, codebreaking will not be necessary. The statistical analysis will be performed blind to patient allocation.

## Intervention group

Participating GPs will review each participant's antihypertensive medication regimen prior to the baseline appointment, and decide which medication should be removed if they are randomised to the intervention arm of the trial. The choice of medication to be withdrawn will be at the discretion of the GP, but their decision will be informed by an individual's comorbidities and existing guidelines, where appropriate (figure 2). Specifically, participating GPs will be encouraged to identify previously unrecognised contraindications to medication, defined by the Screening Tool of Older Person's Prescriptions (STOPP) criteria.[27] In the absence of these, or a strong clinical rationale for continuing despite a STOPP criteria being met, GPs will be recommended to reduce antihypertensive medications in reverse of the National Institute for Health and Care Excellence C+A+D algorithm for older patients (figure 2).[23] All participants in the trial will remain on at least one antihypertensive.

Once a medication has been removed, GPs or other appropriate, delegated healthcare professionals will closely monitor the participant's response to medication reduction: they will be given advice about what and when to monitor (figure 3), but this schedule will be flexible. All participants will be expected to return for at least one routine safety follow-up visit, and further visits may be required if blood pressure is raised (≥150 mm Hg), or AEs occur. Where blood pressure is persistently raised, GPs will be expected to re-adjust medication (dose or type), rendering the likelihood of an SAE occurring as a result of the intervention very low.

**Table 1** Variables and schedule of data collection

| No. | Variable | Data source | | Schedule | |
| --- | --- | --- | --- | --- | --- |
| | | Medical notes | Measured/ collected at clinic | Baseline | Follow-up |
| 1 | Age | | ✓ | ✓ | |
| 2 | Sex | | ✓ | ✓ | |
| 3 | Ethnicity | | ✓ | ✓ | |
| 4 | Marital status | | ✓ | ✓ | |
| 5 | Education | | ✓ | ✓ | |
| 6 | Duration of hypertension | ✓ | | ✓ | |
| 7 | Past medical history | ✓ | | ✓ | |
| 8 | Alcohol consumption | | ✓ | ✓ | ✓ |
| 9 | Smoking | | ✓ | ✓ | ✓ |
| 10 | Height | | ✓ | ✓ | ✓ |
| 11 | Weight | | ✓ | ✓ | ✓ |
| 12 | Clinic blood pressure (sitting and standing) | | ✓ | ✓ | ✓ |
| 13 | Cholesterol (total and HDL) | ✓ | | ✓ | |
| 14 | Estimated glomerular filtration rate | ✓ | | ✓ | |
| 15 | Prescribed or over-the-counter medications (all medications)* | ✓ | ✓ | ✓ | ✓ |
| 16 | Quality of life (according to EQ-5D-5L)[26] | | ✓ | ✓ | ✓ |
| 17 | Functional independence (defined by modified Rankin Scale)[38] | | ✓ | ✓ | |
| 18 | Frailty (according to the FRAIL scale)[39] | | ✓ | ✓ | ✓ |
| 19 | Frailty (according to the frailty index and electronic frailty index)[22 40] | ✓ | ✓ | ✓ | ✓ |
| 20 | Cognitive function (defined by the Montreal Cognitive Assessment)[41] | | ✓ | ✓ | |
| 21 | Adherence to medication (according to the Medication Adherence Rating Scale Questionnaire)[42] | | ✓ | ✓ | ✓ |
| 22 | Adherence to medication reduction | | ✓ | | ✓ |
| 23 | ICD-10 coded cardiovascular events and mortality during the trial | ✓ | | | ✓ |
| 24 | Recording of potential side effects to medication | | ✓ | ✓ | ✓ |
| 25 | Recording of adverse events | ✓ | ✓ | | ✓ |

*Drug substance/name, formulation, dose, frequency, start date and adherence over past 12 months (according to clinical system).
HDL, high-density lipoprotein; ICD, International Statistical Classification of Diseases and Related Health Problems.

## Self-monitoring

All participants randomised to the medication reduction arm of the trial will be given the option to self-monitor their blood pressure at home. Those accepting will be trained using protocols developed in the previous TASMIN trials[28 29] and will be given simple and clear instructions to contact their GP if their blood pressure rises above what is considered clinically safe (eg, *home* systolic blood pressure >145 mm Hg on all readings taken in a week). Participants will be advised to self-monitor (or have a carer monitor) at least four times per week in the last week of each month of follow-up (weeks 4, 8 and 12),

although they can monitor more frequently if they wish. Differential use of self-monitoring in the intervention group, or indeed in the control group (many patients now self-monitor routinely)[30] is not expected to impact on the study results, since there is good evidence that self-monitoring only affects blood pressure levels if used in combination with a co-intervention.[31] All other clinical care will continue as usual.

## Control group

Those allocated to the control arm of the study will continue usual clinical care (ie, they will continue to take

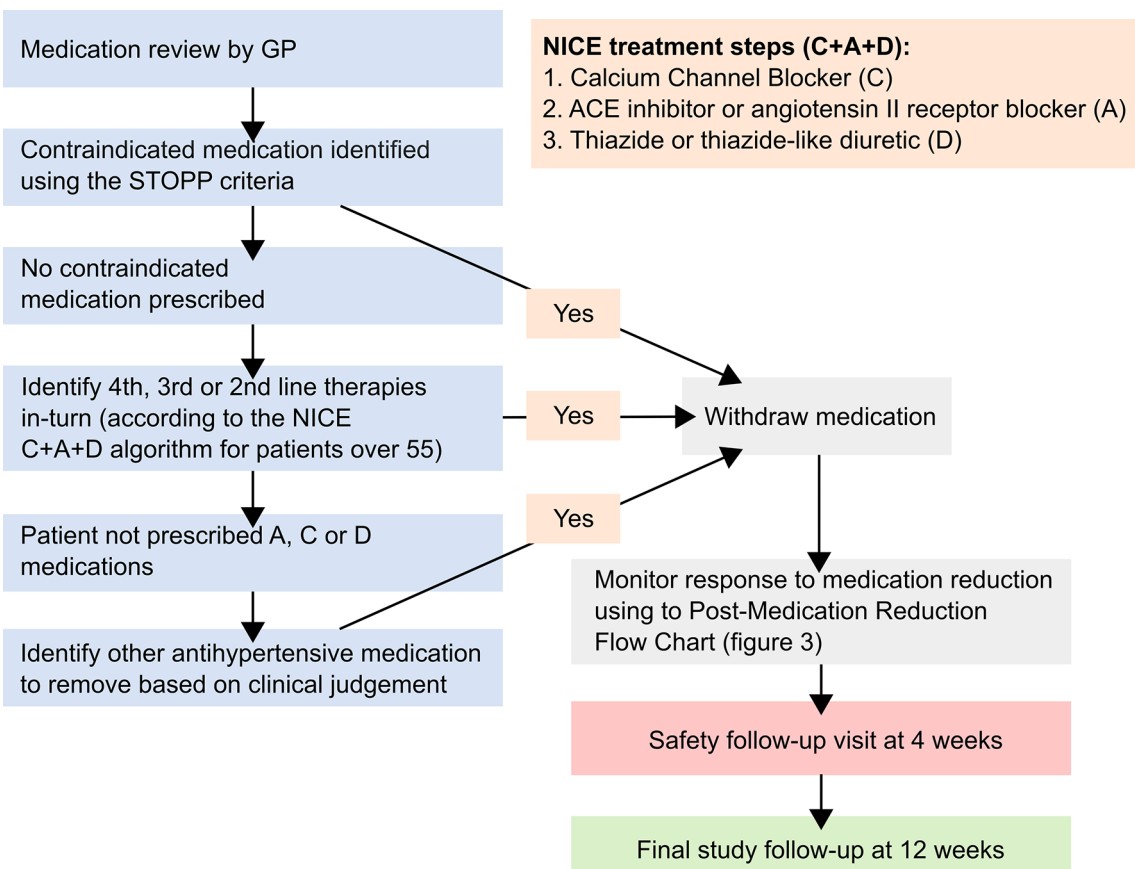

**Figure 2** Medication reduction algorithm. Screening Tool of Older Person's Prescriptions (STOPP) criteria withdraw one of the following medications if any of the ensuing contraindications are identified: thiazide diuretic with a history of gout (may exacerbate gout), beta-blocker in combination with verapamil (risk of symptomatic heart block), non-cardioselective beta-blocker with chronic obstructive pulmonary disease (risk of bronchospasm), calcium channel blockers with chronic constipation (may exacerbate constipation), use of diltiazem or verapamil with New York Heart Association class III or IV heart failure (may worsen heart failure). GP, general practitioner; NICE, National Institute for Health and Care Excellence.

antihypertensive medications as prescribed and will not self-monitor unless already doing so). No other medication changes will be mandated and participating GPs will be asked to manage all other care according to usual clinical practice.

### Follow-up visits

Participants will attend one research follow-up clinic, 12 weeks (±2 weeks) after baseline and those in the intervention will attend one additional safety visit after 4 weeks (±2 weeks) (figure 1). A period of 4 weeks is expected to be sufficiently long enough to assess the impact of antihypertensive medication reduction, since these drugs usually take approximately 4 weeks to 'wash out' of a patient's system. Earlier safety visits are not recommended since they could provide false reassurance that blood pressure is within safe limits if the withdrawn drug has not washed out of the participant's system.

The follow-up assessments will include standardised blood pressure measurement (for assessment of the primary outcome), questionnaire assessments and adherence to the trial medication regime, side effects and AEs (table 1). Where possible, all participants will be flagged for mortality and hospital admissions using National Health Service patient tracking services, permitting long-term follow-up for up to 5 years after the trial has finished.

Each participant has the right to withdraw from the trial at any time. We will ask all participants to attend a follow-up visit as far as is practicable, regardless of whether medication is re-introduced to participants in the intervention group, or a participant in the control group has medication withdrawn. Unless a participant withdraws consent, vital status will be assessed even where an individual has been lost to follow-up (eg, moved away). If given, the reason for withdrawal will be recorded in the eCRF.

### Internal feasibility study

A two-stage internal feasibility study will be conducted to examine methods of patient invitation and rates of recruitment, before proceeding with the main trial. The first feasibility phase will last for a minimum of 3 months and aim to recruit approximately 25 participants from a minimum of 3–5 practices. The aim will be to establish whether or not anyone will be willing to participate in the study.

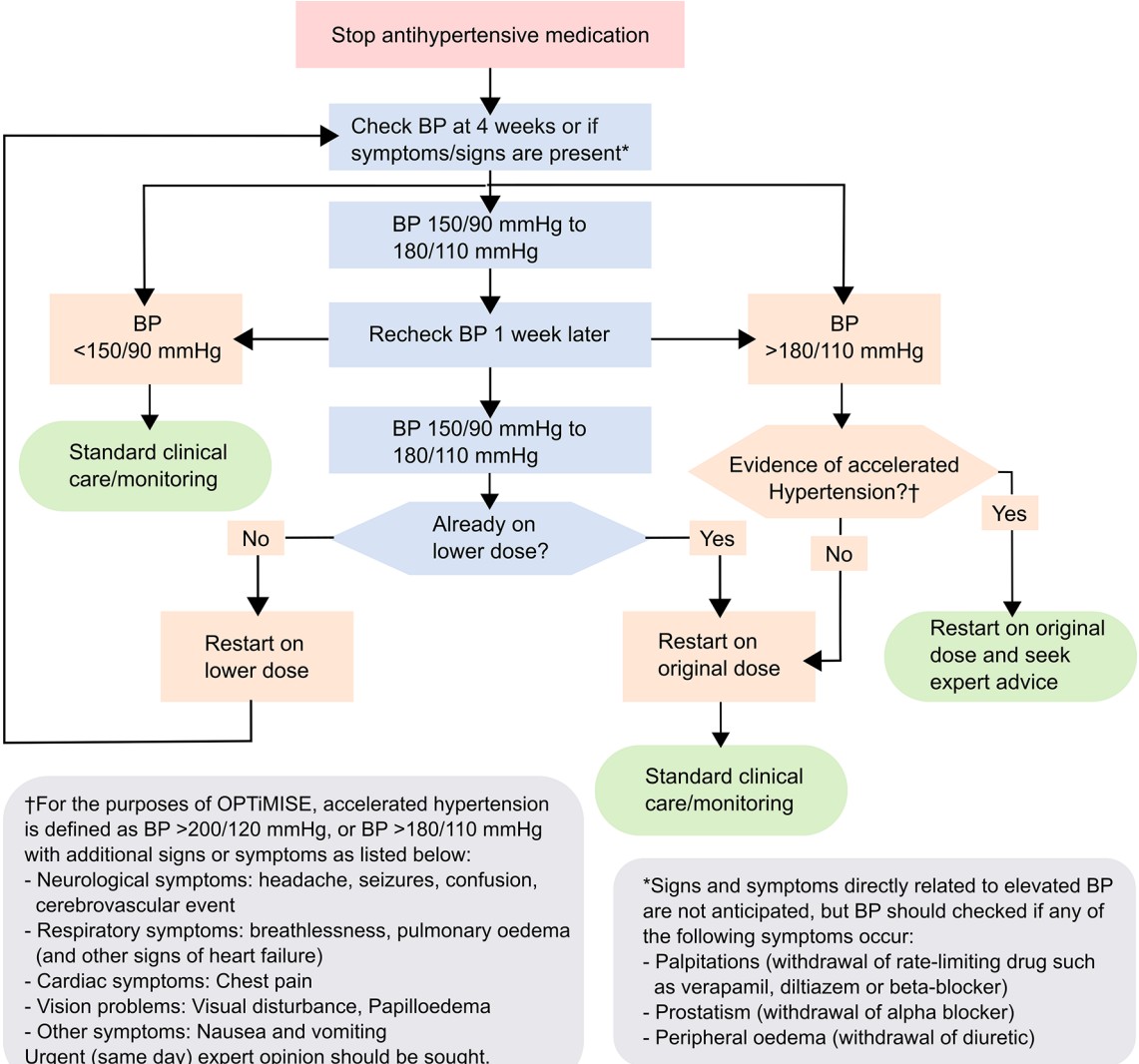

**Figure 3** Postmedication reduction monitoring flow chart. The full effects of most oral antihypertensives can last for up to 4–6 weeks. Frequent monitoring in the initial 4 weeks after drug withdrawal is thus not required unless blood pressure (BP) levels are extreme or there are other clinical concerns (see above). Where systolic/diastolic BP values fall into different categories, consider the higher value. BP should be taken as the averaged second and third measurements using a validated monitor. Standard clinical care/monitoring should align with the National Institute for Health and Care Excellence recommendations.[23]

The second feasibility phase will focus on recruitment rates for the main trial and whether the intended sample size is likely to be met during the recruitment period. This phase will have a recruitment target of 75 participants from 10 practices over 6 months, giving a total sample for the feasibility study of 100 participants. A recruitment rate of 15% of invitations sent is expected. The following actions will be considered to address varying rates of recruitment at the end of the feasibility phases:

► If ≥100 participants are recruited, trial will proceed as planned.
► If 75–99 participants are recruited, recruitment materials/method will be re-examined and edited where necessary following discussions with stakeholders and patient and public involvement representatives.
► If 50–74 participants are recruited, the allocation of resources and recruitment criteria will be re-examined

using information gathered from concurrent qualitative work.
► If <50 participants are recruited, the Trial Steering Committee (TSC) will decide, in discussion with the Data Monitoring and Ethics Committee (DMEC) and the funders, whether the trial should be stopped due to futility.

### Sample size calculation

Assuming that 100% of participants in the usual care group, and 96% of those in the medication reduction group have controlled systolic blood pressure levels (<150 mm Hg) at follow-up, approximately 540 participants will be required to detect a non-inferior difference in systolic blood pressure control between groups. Calculations assume a 10% non-inferiority margin, 90% power, 2.5% one-sided level of significance, 10% loss to follow-up

and a 10% dilution effect due to cross-over between arms. There is no existing precedent for an appropriate margin of non-inferiority in this type of trial, but 10% was deemed useful to inform future doctor-patient discussions about medication reduction: if the non-inferiority margin is met, it will suggest that for every 10 patients who have their medication reduced, at least 9 will still have controlled blood pressure at 12-week follow-up.

## Statistical analysis

A detailed statistical analysis plan will be agreed prior to the end of the trial. The primary and secondary analyses will be by intention to treat (ITT), unless stated otherwise. The primary analysis will be a non-inferiority analysis by means of the 'two one-sided test' procedure,[32] whereby the 95% CI for the relative risk of participants with systolic blood pressure at 12 weeks below 150 mm Hg between the medication reduction group and the usual care group is calculated. This will be obtained by means of a generalised linear mixed effects model with GP surgery included as a random effect and baseline blood pressure as a fixed effect. If the lower limit of the CI is >0.9 (equal to a risk difference of 10%) then the research hypothesis that medication reduction will be non-inferior in terms of blood pressure control to usual care will be accepted. As a secondary analysis of the primary outcome, a per-protocol (PP) analysis will be performed, since ITT can be anticonservative for a non-inferiority hypothesis.[32] Participants who received the medication reduction intervention in the PP analysis will be defined as a participant in the medication reduction arm who maintained their medication reduction throughout the 12-week follow-up period.

Secondary analyses will examine the proportion of participants in the medication reduction arm who maintained their medication reduction throughout the 12-week follow-up period. Secondary outcomes will be analysed by means of linear mixed effects models, adjusting for the baseline level of the outcome and baseline systolic blood pressure and including practice as a random effect: systolic blood pressure, EQ-5D-5L and the Frailty index/frail scale. The difference in the rate of side effects and AEs between the medication reduction and usual care arms will be analysed by means of a logistic mixed effects model adjusting for baseline systolic blood pressure and including practice as a random effect.

Exploratory subgroup analyses of blood pressure control, change in blood pressure and maintenance of medication reduction will be conducted by different levels of baseline frailty, functional independence, cognitive function, number of medications prescribed at baseline and number of comorbidities at baseline.

## Patient and public involvement

This protocol was developed through discussions with older patients and carers and members of an AgeUK day centre. MW is a stroke survivor with experience as a volunteer on the elderly ward of Charing Cross Hospital.

She was consulted on the suitability and design of the trial and is a member of the trial management group. Methods of patient approach, including the design of the video infographic, patient information sheet and consent form were all reviewed by patient representatives prior to formal approval. The TSC includes two independent patient representatives responsible for overseeing the conduct of the trial.

## Safety reporting

AEs that are observed by an investigator or reported by the participant will be recorded on the AE log at any time during the study but AEs will be specifically asked about at the 12-week follow-up. SAEs will be reported to the coordinating centre within 24 hours of discovery or notification of the event. All SAE reports will be reviewed by the DMEC chair on a monthly basis, and by the full DMEC at meetings held every 6 months. The DMEC will include a geriatrician, statistician and consultant clinical pharmacologist. They will be responsible for safeguarding trial participants, monitoring emerging trial data including identifying any trends, such as increases in unexpected events, and take appropriate action where necessary.

All AEs labelled possibly, probably or definitely related will be considered as related to the intervention. Since there are no sections of the Summary of medicinal Product Characteristics, or previous clinical studies which detail expected AEs as a result of medication withdrawal, all SAEs at least possibly related, and not as a result of re-introduction of withdrawn medication, will be considered unexpected and reported as Suspected Unexpected Serious Adverse Reaction (SUSARs). Fatal and life-threatening SUSARs will be reported by the chief investigator to the relevant Competent Authority and Research Ethics Committee no later than 7 calendar days after the sponsor or delegate is first aware of the reaction. All other SUSARs will be reported within 15 calendar days.

## Qualitative substudies
### Study 1: interviews with doctors and patients
Face-to-face interviews with GPs and patients will be conducted to generate understanding about the barriers and facilitators to antihypertensive medication reduction. Informed consent will be sought from approximately 15 GPs to provide a broad range of opinion from varying practice sizes (small to large) and settings (rural to urban). Participating GPs will also be asked to identify up to 15 patients for interview, based on the same inclusion criteria as those applied to participants in the main trial.

Interviews with GPs will use a chart-stimulated recall approach to explore the factors which influence their treatment choices in older hypertensive patients. Anonymised electronic health records will be used to focus discussions about how GPs would feel about reducing antihypertensive medications. Interviews with patients will use 'brown bag' medication review techniques[33] to create a complete record of medication held, with a commentary on usage from the participants' perspective.

Diagrammatic elicitation techniques will be used to complete a relational map outlining participants' circumstances and how these relate to the medications taken. These sketches will be used as the basis for a discussion on the implications of withdrawing antihypertensive medications, and what this 'gap' might mean for the patient.

All interviews will be transcribed verbatim, stored and organised using NVivo software (QSR International, Doncaster, Victoria, Australia). Interview and visual data from GP and patient interviews will be subjected to thematic analysis, with a particular orientation to exploring clinical and patient perspectives on the barriers and facilitators to reducing antihypertensives.

### Study 2: assessment of trial recruitment and data collection procedures

The aim of the second qualitative study will be to explore how information is presented within recruitment appointments and how this might impact on consent to participate, with a view to ensuring robust trial procedures using an iterative process. This will be achieved by audio-recording (with consent) up to 75 consultations between GPs or research assistants and eligible patients.

Thematic analysis will be undertaken on a sample of around 15–20 consultations comprising patients who did, or did not consent to participate, to consider (a) terminology used, (b) presentation of the deprescribing approach and (c) presentation of randomisation. This will inform ongoing trial procedures and future implementation.

### Economic substudy

This work will adapt a previous decision-analytic model examining the long-term costs and benefits from blood pressure-lowering treatment[34] to include potential harms of treatment. The model will be adjusted for the effects of blood pressure lowering on cardiovascular disease risk, costs and quality-adjusted life-years (QALYs) to match the older population involved in this work. Costs of the therapies prescribed, side effects and acute and long-term costs of cardiovascular events will be obtained within the trial and from the literature. Quality of life on each treatment strategy will be obtained from the trial data using EQ-5D-5L, and previous studies will inform utility values for cardiovascular disease health states and the impact of side effects. The model will determine the cost per additional QALY gained of the medication reduction intervention versus usual care and analyses will be conducted from a health and social services perspective. The model will be run with a lifetime perspective, with costs and benefits discounted at a rate of 3.5%. A value of information analysis will assess whether a further trial would be appropriate to reduce decision uncertainty, and identify those parameters where more precise estimates would be most valuable and should therefore be chosen as outcomes for such a trial.

### Ethics and dissemination

This research involves older participants, some of whom may be considered vulnerable. Great care will be taken to ensure all potential participants have the trial clearly explained, and are given sufficient time to decide whether to give informed consent. This will include provision of simplified, participant information sheets with large fonts, video infographics to explain the study and extended GP consultation periods for explaining the study and taking informed consent. The study sponsor has reviewed the study proceedures and ensured all indemnity and insurance requirements for the trial were in place prior to the start of recruitment.

All research outputs from this work will be published in peer-reviewed journals, presented at scientific conferences and lay and social media (eg, Twitter, blogs). 'Patient friendly' study summary documents and infographics will be made available to all participants at the end of the trial via the study website.

### Current trial status

The trial commenced recruitment on 10 April 2017 and is estimated to continue recruitment until September 2018.

### DISCUSSION

Current guidelines in the UK suggest that doctors should ensure that patients are fully informed of the benefits and risks of their prescribed medications and where appropriate, discuss the potential for medication withdrawal in frail individuals with multimorbidity.[35] This is difficult given consultation time constraints and fear that deprescribing might result in harm.[36] This is compounded by conflicting and inconclusive evidence about the benefits and harms of treatment, and a lack of evidence about what will happen if these treatments are reduced.

The ECSTATIC trial enrolled 1067 younger participants aged 40–70 years, taking antihypertensives for primary prevention of cardiovascular disease.[18] The trial demonstrated that only 27% of participants were able to maintain medication reduction throughout follow-up and at 3 months, systolic blood pressure was on average 6 mm Hg higher in the deprescribing group. At 2-year follow-up, the risk of uncontrolled blood pressure was significantly higher in those patients attempting to deprescribe. Unlike the present study, the medication reduction algorithm used did not encourage reintroduction of therapy if blood pressure was persistently raised.

The Discontinuation of Antihypertensive Treatment in Elderly people (DANTE) study[16] examined the effect of complete antihypertensive medication discontinuation in 385 patients over the age of 75 years and with mild cognitive deficits. After 16 weeks of follow-up, they observed a 7/3 mm Hg increase in blood pressure but no difference in overall cognition compound score or quality of life between groups. A study by van der Wardt et al[19] examined the feasibility of a trial reducing antihypertensives in patients with dementia, but was only able to recruit

nine participants for the withdrawal programme (1% recruitment rate) and a larger trial was deemed unfeasible. Similarly, the Opti-Med trial[37] demonstrated in 95 participants that a broader deprescribing approach is achievable in patients living in nursing homes, but was unable to examine the effect on clinical outcomes due to recruitment issues resulting in only 38% of the planned sample size being enrolled.

The OPtimising Treatment for MIld Systolic hypertension in the Elderly (OPTiMISE) trial will target frail individuals with polypharmacy and comorbidity, and aim to establish whether a strategy of antihypertensive medication reduction is safe and acceptable to older patients. The findings of this trial will support better patient-centred management plans for the prevention of cardiovascular disease in older individuals and inform future deprescribing trials in primary care.

**Author affiliations**
[1]Nuffield Department of Primary Care Health Sciences, University of Oxford, Oxford, UK
[2]The Healthcare Improvement Studies Institute, University of Cambridge, Cambridge, UK
[3]Primary Care Research Group, University of Southampton, Southampton, UK
[4]Primary Care Unit, Department of Public Health and Primary Care, University of Cambridge, Cambridge, UK
[5]Institute of Applied Health Research, University of Birmingham, Birmingham, UK
[6]Centre for Academic Primary Care, University of Bristol, Bristol, UK
[7]Patient and Public Involvement Representative, London, UK

**Acknowledgements**  The authors acknowledge the support of the Primary Care Clinical Trials Unit, staff from the NIHR CRNs including Thames Valley and South Midlands, Cambridge, Southampton, West Midlands (Central and South) and West of England and Lucy Curtin for administrative support. Margaret Ogden and Anita Higham serve as patient representatives for the trial steering committee. Additional members of the trial steering committee are Professor Tom Robinson (chair), Professor Rod Taylor and Professor Peter Bower and Professor Richard Lindley. Members of the data monitoring committee are Professor John Gladman (chair), Professor Una Martin and Dr Martyn Lewis. Finally, this work would not be possible without the support of the participating practices and participants.

**Contributors**  JPS conceived, designed and secured funding for the study with JBu, ML, JBe, GAF, CH, FDRH, SJ, PL, JM, EO, RP, MW, L-MY and RJM. JPS wrote the first draft. AN, JMo and L-MY provided the sample size calculations and statistical analysis section. JBu provided the qualitative section. SJ provided the health economic section. All authors reviewed and edited the manuscript. ET is the trial manager. JPS and RJM are co-chief investigators and will act as guarantors for this work.

**Funding**  This work receives joint funding from the National Institute for Health Research (NIHR) Oxford Collaboration for Leadership in Applied Health Research and Care (CLAHRC) at Oxford Health NHS Foundation Trust (ref: P2-501) and the NIHR School for Primary Care Research (SPCR; ref 335). JS and RJMcM have been funded by an NIHR Professorship (NIHR-RP-R2-12-015). FDRH acknowledges part support from the NIHR SPCR, the NIHR CLAHRC Oxford, and the NIHR Oxford Biomedical Research Centre (BRC). CH receives support from the NIHR SPCR and NIHR Oxford BRC. Trial Sponsor: University of Oxford.

**Disclaimer**  The views and opinions expressed are those of the authors and do not necessarily reflect those of the NHS, NIHR or the Department of Health. The sponsor and funder had no role in the study design, writing of the paper; or the decision to submit this protocol for publication, which was made jointly by the authors who have all approved the final manuscript.

**Competing interests**  None declared.

**Patient consent**  Not required.

**Ethics approval**  The protocol, informed consent form, participant information sheet and all other participant facing material have been approved by the

Research Ethics Committee (South Central—Oxford A; ref 16/SC/0628), Medicines and Healthcare products Regulatory Agency (ref 21584/0371/001–0001), host institution(s) and Health Research Authority.

**Provenance and peer review**  Not commissioned; externally peer reviewed.

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
