## [Reviewer comments · BMJ Open]

ARTICLE DETAILS

TITLE (PROVISIONAL)	Optimising Treatment for Mild Systolic hypertension in the Elderly (OPTiMISE): protocol for a randomised controlled non-inferiority trial
AUTHORS	Sheppard, James; Burt, Jenni; Lown, Mark; Temple, Eleanor; Benson, John; Ford, Gary; Heneghan, Carl; Hobbs, Richard; Jowett, Sue; Little, Paul; Mant, Jonathan; Mollison, Jill; Nickless, Alecia; Ogburn, Emma; Payne, Rupert; Williams, Marney; Yu, Ly-Mee; McManus, Richard

VERSION 1 – REVIEW

REVIEWER	Mark Nelson University of Tasmania, AUSTRALIA
REVIEW RETURNED	20-Apr-2018

GENERAL COMMENTS	Optimising Treatment for Mild Systolic hypertension in the Elderly (OPTiMISE): protocol for a randomised controlled non-inferiority trial The investigators are to be congratulated on conducting an interventional trial looking at medication reduction / cessation rather than initiation / intensification. There are too few of the former. An indication as to the quality of the protocol is that as I raised questions or provided suggestions I found later in the document that these had been addressed. Is there a preference for beta-blocker withdrawal given its lower efficacy in reducing stroke in this high risk population (for stroke that is)? No GP selection, NICE treatment regimen does not include beta-blockers for uncomplicated elevated blood pressure but will be found in practice. Would recommend an advisory sheet but free choice for GP. May need step down advice to prevent BP rebound phenomenon with centrally acting agents or beta blockers. GPs may feel they take the risk for adverse events and yet have no guidance. Have a medication reduction algorithm. You will need indemnity insurance approval as this is not routine practice in the absence of drug related adverse effects. Is the preference to reduce number (i.e. one medication to be stopped) or dosage (i.e. one or more to be reduced)? The former will reduce cost and complexity, the latter side-effects (as each agent adverse effects dose related) while retaining efficacy. This seems to be driven by GP choice as patient preference not sought prior to randomisation. Do you have compliance checking of
---

planned GP action prior to randomisation and his/her actions post randomisation?

“A total of 540 participants will be recruited, aged ≥ 80 years, with systolic blood pressure < 150 mmHg and receiving ≥ 2 antihypertensive medications.” The problem with your rationale, the perceived risk of adverse events overwhelming the benefits of MACE reduction is that anecdote may support the former but the evidence the latter. HYVET demonstrated both CVD and SAE benefit (1).

“In an intention-to-treat analysis, active treatment was associated with a 30% reduction in the rate of fatal or nonfatal stroke (95% confidence interval [CI], -1 to 51; $P=0.06$), a 39% reduction in the rate of death from stroke (95% CI, 1 to 62; $P=0.05$), a 21% reduction in the rate of death from any cause (95% CI, 4 to 35; $P=0.02$), a 23% reduction in the rate of death from cardiovascular causes (95% CI, -1 to 40; $P=0.06$), and a 64% reduction in the rate of heart failure (95% CI, 42 to 78; $P<0.001$). Fewer serious adverse events were reported in the active-treatment group (358, vs. 448 in the placebo group; $P=0.001$).”

Safety monitoring extant (hospitalisation due to falls, myocardial infarction, stroke or all-cause mortality).

Non-inferiority design. Needs to ensure that the study is adequately powered to avoid Type 2 error being interpreted as non-inferiority. Is being piloted to inform sample size estimations.

Given open label did you consider cRCT design, i.e. randomisation at the level of the practice to avoid contamination?

A lot of BP lowering studies have WAD as a recruitment strategy as it facilitates recruitment. Have you looked at their literature about their experience of this process rather than just as outcome studies related to yours? We looked at it systematically within ANBP2 (2-4).

“The aim of this study is to determine whether antihypertensive medications can be safely reduced without systolic blood pressure increasing beyond clinically safe levels at follow-up.” The thresholds for safety are arbitrary as SBP is a continuous variable with increasing risk beyond 115 mmHg. Perhaps restate as ‘accepted clinically’.

Incorporates qualitative and economic substudies.

References

1. Beckett NS et al. Treatment of Hypertension in Patients 80 Years of Age or Older. *N Engl J Med* 2008; 358:1887-1898
2. Wing LMH, Reid CM, Ryan P, Beilin LJ, Brown MA, Jennings GLR, et al. A Comparison of Outcomes with Angiotensin-Converting-Enzyme Inhibitors and Diuretics for Hypertension in the Elderly. *N Engl J Med* 2003;348(7): 583-592.

	3. Nelson MR, Reid CM, Muir T, Krum H, Ryan P, McNeil JJ. Predictors of normotension on withdrawal of antihypertensive drugs in second Australian national blood pressure study cohort. BMJ 2002; 325: 815 - 817. 4. Nelson MR, Reid CM, Krum H, Ryan P, Wing LMH, McNeil JJ. Short term predictors of maintenance of normotension post withdrawal of antihypertensive drugs in the Second Australian National Blood Pressure Study (ANBP2). Am J Hypertens; January 2003 16(1) 39-45.
--	--

REVIEWER	Christopher Reid Curtin University/Monash University, Australia
REVIEW RETURNED	23-Apr-2018

GENERAL COMMENTS	The authors have provided a very well described protocol for an antihypertensive drug treatment withdrawal study in frail elderly patients in Primary Care. The rationale for the trial is sound given the potential poly-pharmacy issues in the elderly and the potential to shift the risk-benefit balance. The Inclusion/exclusion criteria for this pragmatic trial is likely to yield a study populations more reflective of those seen in regular clinical practice as opposed to conventional clinical trials. The qualitative sub-studies are an important addition as the likely impact of this trial will rely on GPs, patients and their families being willing to embrace drug withdrawal with the perceived loss of CVD protection associated with withdrawing therapy. The feasibility assessment is also well described and again relates to the community perception of the trial. Questions: 1. Will doctors or nurses be present during the blood pressure recording? 2. The first follow-up visit is 4 weeks after drug withdrawal. The rationale is based on any earlier visit potentially providing "false reassurance that BP is within safe limits. Given many of these patients will have been told for possibly decades that they require these drugs, I would think an earlier follow-up visit (2 weeks) would be of great value for GPs, patients and their families. patients and for the feasibility of the trial.
---

VERSION 1 – AUTHOR RESPONSE

Reviewer(s)' Comments to Author:

Reviewer: 1

The investigators are to be congratulated on conducting an interventional trial looking at medication reduction / cessation rather than initiation / intensification. There are too few of the former. An indication as to the quality of the protocol is that as I raised questions or provided suggestions I found later in the document that these had been addressed.

1. Is there a preference for beta-blocker withdrawal given its lower efficacy in reducing stroke in this high risk population (for stroke that is)? **No, GP selection, NICE treatment regimen does not include beta-blockers for uncomplicated elevated blood pressure but will be found in practice.**

As the reviewer notes, page 6 highlights that the choice of drug to withdraw is left with the GP. In the medication algorithm we suggest that GPs refer to NICE guidelines as part of the decision making process (line 188):

“The choice of medication to be withdrawn will be at the discretion of the GP, but their decision will be informed by an individual’s co-morbidities and existing guidelines, where appropriate (figure 2).”
We have not made any changes to the manuscript.

2. Would recommend an advisory sheet but free choice for GP. May need step down advice to prevent BP rebound phenomenon with centrally acting agents or beta blockers. GPs may feel they take the risk for adverse events and yet have no guidance. Have a medication reduction algorithm.

All GPs are given a medication reduction algorithm (figure 2) and a medication reduction monitoring algorithm (figure 3) as guidance to support the intervention. We have not made any changes to the manuscript.

3. You will need indemnity insurance approval as this is not routine practice in the absence of drug related adverse effects.

The study sponsor has reviewed and ensured all indemnity and insurance requirements for the trial are in place. In the UK, NICE guidelines and CCGs are increasingly encouraging medication reviews and medication reduction as part of routine practice, regardless of whether an individual has suffered adverse events in the past. We have added the following to page 10, line 385:

“The study sponsor reviewed and ensured all indemnity and insurance requirements for the trial were in place prior to the start of recruitment.”

4. Is the preference to reduce number (i.e. one medication to be stopped) or dosage (i.e. one or more to be reduced)? The former will reduce cost and complexity, the latter side-effects (as each agent adverse effects dose related) while retaining efficacy.

The medication reduction intervention involved removing one drug, rather than adjusting the dose of existing drugs. Drugs could be re-introduced at low doses if blood pressure became raised during follow-up. Page 6, line 200 states:

“Where blood pressure is persistently raised, GPs will be expected to re-adjust medication (dose or type), rendering the likelihood of a serious adverse event occurring as a result of the intervention very low.”

5. This seems to be driven by GP choice as patient preference not sought prior to randomisation. Do you have compliance checking of planned GP action prior to randomisation and his/her actions post randomisation?

This is correct. Once the patient has been consented, they are seen by a research facilitator/nurse who collects the baseline data and randomises the patient. They inform patients randomised to the intervention of the GP’s choice of drug to withdraw. We have clarified this in the methods on page 5, line 157:

“Those giving informed consent will be screened using the criteria in table 1 and undergo baseline measurements and randomisation by a member of the research team via participant questionnaires and a detailed notes review (table 2).”

We will compare the drug listed in the GPs medication reduction plan to drug remove following randomisation as part of the analysis.

6. “A total of 540 participants will be recruited, aged ≥ 80 years, with systolic blood pressure < 150 mmHg and receiving ≥ 2 antihypertensive medications.” The problem with your rationale, the perceived risk of adverse events overwhelming the benefits of MACE reduction is that anecdote may support the former but the evidence the latter. HYVET demonstrated both CVD and SAE benefit (1).

“In an intention-to-treat analysis, active treatment was associated with a 30% reduction in the rate of fatal or nonfatal stroke (95% confidence interval [CI], -1 to 51; P=0.06), a 39% reduction in the rate of death from stroke (95% CI, 1 to 62; P=0.05), a 21% reduction in the rate of death from any cause (95% CI, 4 to 35; P=0.02), a 23% reduction in the rate of death from cardiovascular causes (95% CI, -1 to 40; P=0.06), and a 64% reduction in the rate of heart failure (95% CI, 42 to 78; P<0.001). Fewer serious adverse events were reported in the active-treatment group (358, vs. 448 in the placebo group; P=0.001).”

Safety monitoring extant (hospitalisation due to falls, myocardial infarction, stroke or all-cause mortality).

We thank the reviewer for highlighting this. Our rationale suggests that the potential harms of treatment may outweigh the benefits in frail, multi-morbid patients not usually enrolled into randomised controlled trials. It is of course true the some older patients will still benefit from BP lowering as they did in the HYVET trial. However, our eligibility criteria would preclude anyone eligible for HYVET from participating in this trial – specifically we only enrol patients with a baseline BP <150 mmHg systolic whereas patients were only eligible for HYVET if they had a baseline BP >160 mmHg. We have not made any changes to the manuscript.

7. Non-inferiority design. Needs to ensure that the study is adequately powered to avoid Type 2 error being interpreted as non-inferiority. Is being piloted to inform sample size estimations.

We agree and the study sample size takes this into account. It is described on page 8, line 263: *“Assuming that 100% of participants in the usual care group, and 96% of those in the medication reduction group have controlled systolic blood pressure levels (<150mmHg) at follow-up, approximately 540 participants will be required to detect a non-inferior difference in systolic blood pressure control between groups. Calculations assume a 10% non-inferiority margin, 90% power, 2.5% 1-sided level of significance, 10% loss to follow-up and a 10% dilution effect due to cross-over between arms. There is no existing precedent for an appropriate margin of non-inferiority in this type of trial, but 10% was deemed useful to inform future doctor-patient discussions about medication reduction: if the non-inferiority margin is met, it will suggest that for every ten patients who have their medication reduced, at least nine will still have controlled blood pressure at 12 week follow-up.”*

8. Given open label did you consider cRCT design, i.e. randomisation at the level of the practice to avoid contamination?

We chose not to cluster randomise the study, but do stratify the randomisation by practice to reduce the chances of contamination. We describe this on page 6, line 177:

“A computer generated non-deterministic algorithm, minimising on practice and baseline systolic blood pressure will be used to ensure these covariates are balanced between the two intervention arms.”

9. A lot of BP lowering studies have WAD as a recruitment strategy as it facilitates recruitment. Have you looked at their literature about their experience of this process rather than just as outcome studies related to yours? We looked at it systematically within ANBP2 (2-4).

We thank the reviewer for this suggestion. We were aware that such an approach has been used in previous trials (and note that this was used in HYVET in the introduction) but had not seen these useful references from the ANBP2 trial. We have included them in the introduction on page 4, line 91:

“The HYVET trial enrolled some patients on antihypertensive treatment who were then randomised to placebo (effectively complete medication withdrawal) and the ANBP2 trial investigators followed up participants who withdrew medication during the trial run-in period but who were not randomised into the trial. They found younger patients with lower baseline blood pressure and fewer drug prescriptions were more likely to sustain medication withdrawal at 12 month follow-up.”

10. “The aim of this study is to determine whether antihypertensive medications can be safely reduced without systolic blood pressure increasing beyond clinically safe levels at follow-up.” The thresholds for safety are arbitrary as SBP is a continuous variable with increasing risk beyond 115 mmHg. Perhaps restate as ‘accepted clinically’.

We agree and modified accordingly on page 4, line 108:

“The aim of this study is to determine whether antihypertensive medications can be safely reduced without systolic blood pressure increasing beyond what is clinical acceptable at follow-up.”

Reviewer: 2

The authors have provided a very well described protocol for an antihypertensive drug treatment withdrawal study in frail elderly patients in Primary Care. The rationale for the trial is sound given the potential poly-pharmacy issues in the elderly and the potential to shift the risk-benefit balance. The Inclusion/exclusion criteria for this pragmatic trial is likely to yield a study populations more reflective of those seen in regular clinical practice as opposed to conventional clinical trials. The qualitative sub-studies are an important addition as the likely impact of this trial will rely on GPs, patients and their families being willing to embrace drug withdrawal with the perceived loss of CVD protection associated with withdrawing therapy. The feasibility assessment is also well described and again relates to the community perception of the trial.

1. Will doctors or nurses be present during the blood pressure recording?

No. this will be undertaken by the research facilitator or research nurse. We have clarified this on page 6, line 165:

“Only the research facilitator/nurse will be present during the blood pressure measurements.”

2. The first follow-up visit is 4 weeks after drug withdrawal. The rationale is based on any earlier visit potentially providing “false re-assurance that BP is within safe limits.” Given many of these patients will have been told for possibly decades that they require these drugs, I would think an earlier follow-up visit (2 weeks) would be of great value for GPs, patients and their families. patients and for the feasibility of the trial.

We thank the reviewer for this suggestion. The period between baseline and the first safety follow-up visit was discussed at length by the research team during the planning of this trial and a period of 4 weeks was chosen for the reasons given above. We do also offer patients the opportunity to self-monitor their blood pressure throughout follow-up so those wishing for reassurance can take this option. As part of this self-monitoring, participants are given clear instructions on when to contact the surgery regarding raised BP readings. At this stage it is not possible to make changes to the follow-up schedule so we have not made any changes to the manuscript.

VERSION 2 – REVIEW

REVIEWER	Christopher Reid Curtin University / Monash University, Australia
REVIEW RETURNED	25-Jun-2018
GENERAL COMMENTS	Revision has addressed issues raised